# Spike train entropy-rate estimation using hierarchical Dirichlet process priors

**Karin Knudson**
Department of Mathematics
kknudson@math.utexas.edu

**Jonathan W. Pillow**
Center for Perceptual Systems
Departments of Psychology & Neuroscience
The University of Texas at Austin
pillow@mail.utexas.edu

## Abstract

Entropy rate quantifies the amount of disorder in a stochastic process. For spiking neurons, the entropy rate places an upper bound on the rate at which the spike train can convey stimulus information, and a large literature has focused on the problem of estimating entropy rate from spike train data. Here we present Bayes least squares and empirical Bayesian entropy rate estimators for binary spike trains using hierarchical Dirichlet process (HDP) priors. Our estimator leverages the fact that the entropy rate of an ergodic Markov Chain with known transition probabilities can be calculated analytically, and many stochastic processes that are non-Markovian can still be well approximated by Markov processes of sufficient depth. Choosing an appropriate depth of Markov model presents challenges due to possibly long time dependencies and short data sequences: a deeper model can better account for long time dependencies, but is more difficult to infer from limited data. Our approach mitigates this difficulty by using a hierarchical prior to share statistical power across Markov chains of different depths. We present both a fully Bayesian and empirical Bayes entropy rate estimator based on this model, and demonstrate their performance on simulated and real neural spike train data.

## 1   Introduction

The problem of characterizing the statistical properties of a spiking neuron is quite general, but two interesting questions one might ask are: (1) what kind of time dependencies are present? and (2) how much information is the neuron transmitting? With regard to the second question, information theory provides quantifications of the amount of information transmitted by a signal without reference to assumptions about how the information is represented or used. The entropy rate is of interest as a measure of uncertainty per unit time, an upper bound on the rate of information transmission, and an intermediate step in computing mutual information rate between stimulus and neural response.

Unfortunately, accurate entropy rate estimation is difficult, and estimates from limited data are often severely biased. We present a Bayesian method for estimating entropy rates from binary data that uses hierarchical Dirichlet process priors (HDP) to reduce this bias. Our method proceeds by modeling the source of the data as a Markov chain, and then using the fact that the entropy rate of a Markov chain is a deterministic function of its transition probabilities. Fitting the model yields parameters relevant to both questions (1) and (2) above: we obtain both an approximation of the underlying stochastic process as a Markov chain, and an estimate of the entropy rate of the process.

For binary data, the HDP reduces to a hierarchy of beta priors, where the prior probability over $g$, the probability of the next symbol given a long history, is a beta distribution centered on the probability of that symbol given a truncated, one-symbol-shorter, history. The posterior over symbols given

a certain history is thus "smoothed" by the probability over symbols given a shorter history. This smoothing is a key feature of the model.

The structure of the paper is as follows. In Section 2, we present definitions and challenges involved in entropy rate estimation, and discuss existing estimators. In Section 3, we discuss Markov models and their relationship to entropy rate. In Sections 4 and 5, we present two Bayesian estimates of entropy rate using the HDP prior, one involving a direct calculation of the posterior mean transition probabilities of a Markov model, the other using Markov Chain Monte Carlo methods to sample from the posterior distribution of the entropy rate. In Section 6 we compare the HDP entropy rate estimators to existing entropy rate estimators including the context tree weighting entropy rate estimator from [1], the string-parsing method from [2], and finite-length block entropy rate estimators that makes use of the entropy estimator of Nemenman, Bialek and Shafee [3] and Miller and Madow [4]. We evaluate the results for simulated and real neural data.

## 2 Entropy Rate Estimation

In information theory, the entropy of a random variable is a measure of the variable's average unpredictability. The entropy of a discrete random variable $X$ with possible values $\{x_1, ..., x_n\}$ is

$$H(X) = -\sum_{i=1}^{n} p(x_i) \log(x_i) \tag{1}$$

Entropy can be measured in either nats or bits, depending on whether we use base 2 or $e$ for the logarithm. Here, all logarithms discussed will be base 2, and all entropies will be given in bits.

While entropy is a property of a random variable, entropy *rate* is a property of a stochastic process, such as a time series, and quantifies the amount of uncertainty *per symbol*. The neural and simulated data considered here will be binary sequences representing the spike train of a neuron, where each symbol represents either the presence of a spike in a bin (1) or the absence of a spike (0). We view the data as a sample path from an underlying stochastic process. To evaluate the average uncertainty of each new symbol (0 or 1) given the previous symbols - or the amount of new information per symbol - we would like to compute the entropy rate of the process.

For a stochastic process $\{X_i\}_{i=1}^{\infty}$ the entropy of the random vector $(X_1, ..., X_k)$ grows with $k$; we are interested in how it grows. If we define the block entropy $H_k$ to be the entropy of the distribution of length-$k$ sequences of symbols, $H_k = H(X_{i+1}, ...X_{i+k})$, then the entropy rate of a stochastic process $\{X_i\}_{i=1}^{\infty}$ is defined by

$$h = \lim_{k \to \infty} \frac{1}{k} H_k \tag{2}$$

when the limit exists (which, for stationary stochastic processes, it must). There are two other definitions for entropy rate, which are equivalent to the first for stationary processes:

$$h = \lim_{k \to \infty} H_{k+1} - H_k \tag{3}$$

$$h = \lim_{k \to \infty} H(X_{i+1}|X_i, X_{i-1}, ...X_{i-k}) \tag{4}$$

We now briefly review existing entropy rate estimators, to which we will compare our results.

### 2.1 Block Entropy Rate Estimators

Since much work has been done to accurately estimate entropy from data, Equations (2) and (3) suggest a simple entropy rate estimator, which consists of choosing first a block size $k$ and then a suitable entropy estimator with which to estimate $H_k$. A simple such estimator is the "plugin" entropy estimator, which approximates the probability of each length-$k$ block $(x_1, ..., x_k)$ by the proportion of total length-$k$ blocks observed that are equal to $(x_1, ..., x_k)$. For binary data there are

$2^k$ possible length-$k$ blocks. When $N$ denotes the data length and $c_i$ the number of observations of each block in the data, we have:

$$\hat{H}_{\text{plugin}} = \sum_{i=1}^{2^k} -\frac{c_i}{N} \log \frac{c_i}{N} \qquad (5)$$

from which we can immediately estimate the entropy rate with $h_{plugin,k} = \hat{H}_{\text{plugin}}/k$, for some appropriately chosen $k$ (the subject of "appropriate choice" will be taken up in more detail later).

We would expect that using better block entropy estimators would yield better entropy rate estimators, and so we also consider two other block based entropy rate estimators. The first uses the Bayesian entropy estimator $H_{NSB}$ from Nemenman, Shafee and Bialek [3], which gives a Bayesian least squares estimate for entropy given a mixture-of-Dirichlet prior. The second uses the Miller and Madow estimator [4], which gives a first-order correction to the (often significantly biased) plugin entropy estimator of Equation 5:

$$\hat{H}_{MM} = \sum_{i=1}^{2^k} -\frac{c_i}{N} \log \frac{c_i}{N} + \frac{A-1}{2N} \log(e) \qquad (6)$$

where $A$ is the size of the alphabet of symbols ($A = 2$ for the binary data sequences presently considered). For a given $k$, we obtain entropy rate estimators $h_{NSB,k} = \hat{H}_{NSB}/k$ and $h_{MM,k} = \hat{H}_{MM}/k$ by applying the entropy estimators from [3] and [4] respectively to the empirical distribution of the length-$k$ blocks.

While we can improve the accuracy of these block entropy rate estimates by choosing a better entropy estimator, choosing the block size $k$ remains a challenge. If we choose $k$ to be small, we miss long time dependencies in the data and tend to overestimate the entropy; intuitively, the time series will seem more unpredictable than it actually is, because we are ignoring long-time dependencies. On the other hand, as we consider larger $k$, limited data leads to underestimates of the entropy rate. See the plots of $h_{plugin}$, $h_{NSB}$, and $h_{MM}$ in Figure 2d for an instance of this effect of block size on entropy rate estimates. We might hope that in between the overestimates of entropy rate for short blocks and the the underestimates for longer blocks, there is some "plateau" region where the entropy rate stays relatively constant with respect to block size, which we could use as a heuristic to select the proper block length [1]. Unfortunately, the block entropy rate at this plateau may still be biased, and for data sequences that are short with respect to their time dependencies, there may be no discernible plateau at all ([1], Figure 1).

## 2.2    Other Entropy Rate Estimators

Not all existing techniques for entropy rate estimation involve an explicit choice of block length. The estimator from [2], for example, parses the full string of symbols in the data by starting from the first symbol, and sequentially removing and counting as a "phrase" the shortest substring that has not yet appeared. When $M$ is the number of distinct phrases counted in this way, we obtain the estimator: $h_{LZ} = \frac{M}{N} \log N$, free from any explicit block length parameters.

A fixed block length model like the ones described in the previous section uses the entropy of the distribution of all the blocks of a some length - e.g. all the blocks in the terminal nodes of a context tree like the one in Figure 1a. In the context tree weighting (CTW) framework of [1], the authors instead use a minimum descriptive length criterion to weight different tree topologies, which have within the same tree terminal nodes corresponding to blocks of different lengths. They use this weighting to generate Monte Carlo samples and approximate the integral $\int h(\theta)p(\theta|\text{T}, \text{data})p(\text{T}|\text{data}) \, d\theta \, d\text{T}$, in which T represents the tree topology, and $\theta$ represents transition probabilities associated with the terminal nodes of the tree.

In our approach, the HDP prior combined with a Markov model of our data will be a key tool in overcoming some of the difficulties of choosing a block-length appropriately for entropy rate estimation. It will allow us to choose a block length that is large enough to capture possibly important long time dependencies, while easing the difficulty of estimating the properties of these long time dependencies from short data.

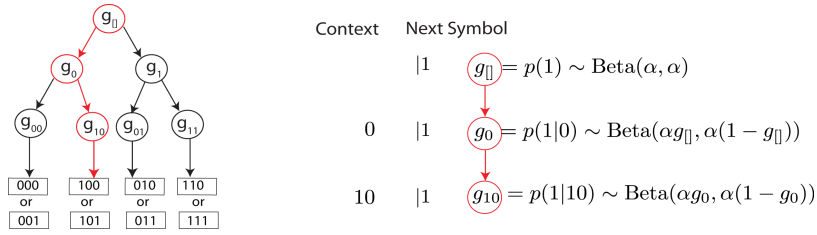

Figure 1: A depth-3 hierarchical Dirichlet prior for binary data

## 3 Markov Models

The usefulness of approximating our data source with a Markov model comes from (1) the flexibility of Markov models including their ability to well approximate even many processes that are not truly Markovian, and (2) the fact that for a Markov chain with known transition probabilities the entropy rate need not be estimated but is in fact a deterministic function of the transition probabilities.

A Markov chain is a sequence of random variables that has the property that the probability of the next state depends only on the present state, and not on any previous states. That is, $P(X_{i+1}|X_i, ..., X_1) = P(X_{i+1}|X_i)$. Note that this property does not mean that for a binary sequence the probability of each 0 or 1 depends only on the previous 0 or 1, because we consider the state variables to be strings of symbols of length $k$ rather than individual 0s and 1s, Thus we will discuss "depth-$k$" Markov models, where the probability of the next state depends only previous $k$ symbols, or what we will call the length-$k$ *context* of the symbol. With a binary alphabet, there are $2^k$ states the chain can take, and from each state $s$, transitions are possible only to two other states. (So that for, example, 1**10** can transition to state **10**1 or state **10**0, but not to any other state). Because only two transitions are possible from each state, the transition probability distribution from each $s$ is completely specified by only one parameter, which we denote $g_s$, the probability of observing a 1 given the context $s$.

The entropy rate of an ergodic Markov chain with finite state set $A$ is given by:

$$h = \sum_{s \in A} p(s)H(x|s), \tag{7}$$

where $p(s)$ is the stationary probability associated with state $s$, and $H(x|s)$ is the entropy of the distribution of possible transitions from state $s$. The vector of stationary state probabilities $p(\mathbf{s})$ for all $s$ is computed as a left eigenvector of the transition matrix $\mathbf{T}$:

$$p(\mathbf{s})\mathbf{T} = p(\mathbf{s}) \ , \ \sum_s p(s) = 1 \tag{8}$$

Since each row of the transition matrix $\mathbf{T}$ contains only two non-zero entries, $g_s$, and $1 - g_s, p(\mathbf{s})$ can be calculated relatively quickly. With equations 7 and 8, $h$ can be calculated analytically from the vector of all $2^k$ transition probabilities $\{g_s\}$. A Bayesian estimator of entropy rate based on a Markov model of order $k$ is given by

$$\hat{h}_{Bayes} = \int h(\mathbf{g})p(\mathbf{g}|\text{data})d\mathbf{g} \tag{9}$$

where $\mathbf{g} = \{g_s : |s| = k\}$, $h$ is the deterministic function of $\mathbf{g}$ given by Equations 7 and 8, and $p(\mathbf{g}|\text{data}) \propto p(\text{data}|\mathbf{g})p(\mathbf{g})$ given some appropriate prior over $\mathbf{g}$.

Modeling a time series as a Markov chain requires a choice of the depth of that chain, so we have not avoided the depth selection problem yet. What will actually mitigate the difficulty here is the use of hierarchical Dirichlet process priors.

## 4 Hierarchical Dirichlet Process priors

We describe a hierarchical beta prior, a special case of the hierarchical Dirichlet process (HDP), which was presented in [5] and applied to problems of natural language processing in [6] and [7].

The true entropy rate $h = \lim_{k \to \infty} H_k/k$ captures time dependencies of infinite depth. Therefore to calculate the estimate $\hat{h}_{Bayes}$ in Equation 9 we would like to choose some large $k$. However, it is difficult to estimate transition probabilities for long blocks with short data sequences, so choosing large $k$ may lead to inaccurate posterior estimates for the transition probabilities $\mathbf{g}$. In particular, shorter data sequences may not even have observations of all possible symbol sequences of a given length.

This motivates our use of hierarchical priors as follows. Suppose we have a data sequence in which the subsequence 0011 is never observed. Then we would not expect to have a very good estimate for $g_{0011}$; however, we could improve this by using the assumption that, a priori, $g_{0011}$ should be similar to $g_{011}$. That is, the probability of observing a 1 after the context sequence 0011 should be similar to that of seeing a 1 after 011, since it might be reasonable to assume that context symbols from the more distant past matter less. Thus we choose for our prior:

$$g_s | g_{s'} \sim \text{Beta}(\alpha_{|s|} g_{s'}, \alpha_{|s|}(1 - g_{s'})) \qquad (10)$$

where $s'$ denotes the context $s$ with the earliest symbol removed. This choice gives the prior distribution of $g_s$ mean $g_{s'}$, as desired. We continue constructing the prior with $g_{s''} | g_{s'} \sim \text{Beta}(\alpha_{|s'|} g_{s''}, \alpha_{|s'|}(1 - g_{s''}))$ and so on until $g_{[]} \sim \text{Beta}(\alpha_0 p_\emptyset, \alpha_0(1 - p_\emptyset))$ where $g_{[]}$ is the probability of a spike given no context information and $p_\emptyset$ is a hyperparameter reflecting our prior belief about the probability of a spike. This hierarchy gives our prior the tree structure as shown in in Figure 1. A priori, the distribution of each transition probability is centered around the transition probability from a one-symbol-shorter block of symbols. As long as the assumption that more distant contextual symbols matter less actually holds (at least to some degree), this structure allows the sharing of statistical information across different contextual depths. We can obtain reasonable estimates for the transition probabilities from long blocks of symbols, even from data that is so short that we may have few (or no) observations of each of these long blocks of symbols.

We could use any number of distributions with mean $g_{s'}$ to center the prior distribution of $g_s$ at $g_{s'}$; we use Beta distributions because they are conjugate to the likelihood. The $\alpha_{|s|}$ are concentration parameters which control how concentrated the distribution is about its mean, and can also be estimated from the data. We assume that there is one value of $\alpha$ for each level in the hierarchy, but one could also fix alpha to be constant throughout all levels, or let it vary within each level.

This hierarchy of beta distributions is a special case of the hierarchical Dirichlet process . A Dirichlet process (DP) is a stochastic process whose sample paths are each probability distributions. Formally, if $G$ is a finite measure on a set S, then $X \sim DP(\alpha, G)$ if for any finite measurable partition of the sample space $(A_1, ... A_n)$ we have that $X(A_1), ... X(A_n) \sim \text{Dirichlet}(\alpha G(A_1), ..., \alpha G(A_n))$. Thus for a partition into only two sets, the Dirichlet process reduces to a beta distribution, which is why when we specialize the HDP to binary data, we obtain a hierarchical beta distribution. In [5] the authors present a hierarchy of DPs where the base measure for each DP is again a DP. In our case, for example, we have $G_{011} = \{g_{011}, 1 - g_{011}\} \sim DP(\alpha_3, G_{11})$, or more generally, $G_s \sim DP(\alpha_{|s|}, G_{s'})$.

## 5    Empirical Bayesian Estimator

One can generate a sequence from an HDP by drawing each subsequent symbol from the transition probability distribution associated with its context, which is given recursively by [6] :

$$p(1|s) = \begin{cases} \frac{c_{s1}}{\alpha_{|s|} + c_s} + \frac{\alpha_{|s|}}{\alpha_{|s|} + c_s} p(1|s') & \text{if } s \neq \emptyset \\ \frac{c_1}{\alpha_0 + N} + \frac{\alpha_0}{\alpha_0 + N} p_\emptyset & \text{if } s = \emptyset \end{cases} \qquad (11)$$

where $N$ is the length of the data string, $p_\emptyset$ is a hyperparameter representing the a prior probability of observing a 1 given no contextual information, $c_{s1}$ is the number of times the symbol sequence $s$ followed by a 1 was observed, and $c_s$ is the number of times the symbol sequence s was observed.

We can calculate the posterior predictive distribution $\hat{\mathbf{g}}_{pr}$ which is specified by the $2^k$ values $\{g_s = p(1|s) : |s| = k\}$ by using counts $c$ from the data and performing the above recursive calculation to estimate $g_s$ for each of the $2^k$ states $s$. Given the estimated Markov transition probabilities $\hat{\mathbf{g}}_{pr}$ we then have an empirical Bayesian entropy rate estimate via Equations 7 and 8. We denote this estimator $h_{empHDP}$. Note that while $\hat{\mathbf{g}}_{pr}$ is the posterior mean of the transition probabilities, the

entropy rate estimator $h_{empHDP}$ is no longer a fully Bayesian estimate, and is not equivalent to the $\hat{h}_{Bayes}$ of equation 9. We thus lose some clarity and the ability to easily compute Bayesian confidence intervals. However, we gain a good deal of computational efficiency because calculating $h_{empHDP}$ from $\hat{\mathbf{g}}_{pr}$ involves only one eigenvector computation, instead of the many needed for the MC approximation to the integral in Equation 9. We present a fully Bayesian estimate next.

## 6 Fully Bayesian Estimator

Here we return to the Bayes least squares estimator $\hat{h}_{Bayes}$ of Equation 9. The integral is not analytically tractable, but we can approximate it using Markov Chain Monte Carlo techniques. We use Gibbs sampling to simulate $N_{MC}$ samples $\mathbf{g}^{(i)} \sim \mathbf{g}|data$ from the posterior distribution and then calculate $h^{(i)}$ from each $\mathbf{g}^{(i)}$ via Equations 7 and 8 to obtain the Bayesian estimate:

$$h_{HDP} = \frac{1}{N_{MC}} \sum_{i=1}^{N_{MC}} h^{(i)} \tag{12}$$

To perform the Gibbs sampling, we need the posterior conditional probabilities of each $g_s$. Because the parameters of the model have the structure of a tree, each $g_s$ for $|s| < k$ is conditionally independent from all but its immediate ancestor in the tree, $g_{s'}$, and its two descendants, $g_{0s}$ and $g_{1s}$. We have:

$$p(g_s|g_{s'}, g_{0s}, g_{1s}.\alpha_{|s|}, \alpha_{|s|=1}) \propto \text{Beta}(g_s; \alpha_{|s|}g_{s'}, \alpha_{|s|}(1 - g_{s'}))\text{Beta}(g_{0s}; \alpha_{|s|+1}g_s, \alpha_{|s|+1}(1 - g_s))$$
$$\text{Beta}(g_{1s}; \alpha_{|s|+1}g_s, \alpha_{|s|+1}(1 - g_s)) \tag{13}$$

and we can compute these probabilities on a discrete grid since they are each one dimensional, then sample the posterior $g_s$ via this grid. We used a uniform grid of 100 points on the interval [0,1] for our computation. For the transition probabilities from the bottom level of the tree $\{g_s : |s| = k\}$, the conjugacy of the beta distributions with binomial likelihood function gives the posterior conditional of $g_s$ a recognizable form: $p(g_s|g_{s'}, \text{data}) = \text{Beta}(\alpha_k g_{s'} + c_{s1}, \alpha_k(1 - g_{s'}) + c_{s0})$.

In the HDP model we may treat each $\alpha$ as a fixed hyperparameter, but it is also straightforward to set a prior over each $\alpha$ and then sample $\alpha$ along with the other model parameters with each pass of the Gibbs sampler. The full posterior conditional for $\alpha_i$ with a uniform prior is (from Bayes' theorem):

$$p(\alpha_i|g_s, g_{s0}, g_{s1} : |s| = i - 1) \propto \prod_{\{s:|s|=i-1\}} \frac{(g_{s1}g_{s0})^{\alpha_i g_s - 1}((1 - g_{s1})(1 - g_{s0}))^{\alpha_i(1-g_s)-1}}{\text{Beta}(\alpha_i g_s, \alpha_i(1 - g_s))^2} \tag{14}$$

We sampled $\alpha$ by computing the probabilities above on a grid of values spanning the range $[1, 2000]$. This upper bound on $\alpha$ is rather arbitrary, but we verified that increasing the range for $\alpha$ had little effect on the entropy rate estimate, at least for the ranges and block sizes considered.

In some applications, the Markov transition probabilities $\mathbf{g}$, and not just the entropy rate, may be of interest as a description of the time dependencies present in the data. The Gibbs sampler above yields samples from the distribution $\mathbf{g}|data$, and averaging these $N_{MC}$ samples yields a Bayes least squares estimator of transition probabilities, $\hat{\mathbf{g}}_{gibbsHDP}$. Note that this estimate is closely related to the estimate $\hat{\mathbf{g}}_{pr}$ from the previous section; with more MC samples, $\hat{\mathbf{g}}_{gibbsHDP}$ converges to the posterior mean $\hat{\mathbf{g}}_{pr}$ (when the $\alpha$ are fixed rather than sampled, to match the fixed $\alpha$ per level used in Equation 11).

## 7 Results

We applied the model to both simulated data with a known entropy rate and to neural data, where the entropy rate is unknown. We examine the accuracy of the fully Bayesian and empirical Bayesian entropy rate estimators $h_{HDP}$ and $h_{empHDP}$, and compare the entropy rate estimators $h_{plugin}$, $h_{NSB}$, $h_{MM}$, $h_{LZ}$ [2], and $h_{CTW}$ [1], which are described in Section 2. We also consider estimates of the Markov transition probabilities $\mathbf{g}$ produced by both inference methods.

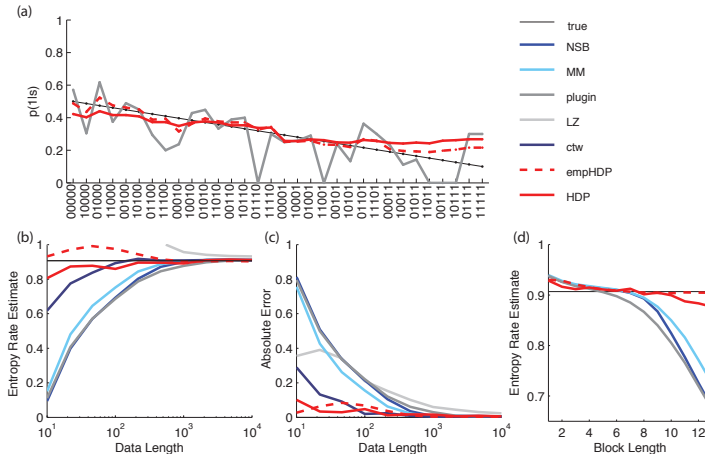

Figure 2: Comparison of estimated (a) transition probability and (b,c,d) entropy rate for data simulated from a Markov model of depth 5. In (a) and (d), data sets are 500 symbols long. The block-based and HDP estimators in (b) and (c) use block size $k = 8$. In (b,c,d) results were averaged over 5 data sequences, and (c) plots the average absolute value of the difference between true and estimated entropy rates.

## 7.1 Simulation

We considered data simulated from a Markov model with transition probabilities set so that transition probabilities from states with similar suffixes are similar (i.e. the process actually does have the property that more distant context symbols matter less than more recent ones in determining transitions). We used a depth-5 Markov model, whose true transition probabilities are shown in black in Figure 2a , where each of the 32 points on the x axis represents the probability that the next symbol is a 1 given the specified 5-symbol context.

In Figure 2a we compare HDP estimates of transition probabilities of this simulated data to the plugin estimator of transition probabilities $\hat{g}_s = \frac{c_{s1}}{c_s}$ calculated from a 500-symbol sequence. (The other estimators do not include calculating transition probabilities as an intermediate step, and so cannot be included here.) With a series of 500 symbols, we do not expect enough observations of each of possible transitions to adequately estimate the $2^k$ transition probabilities, even for rather modest depths such as $k = 5$. And indeed, the "plugin" estimates of transition probabilities do not match the true transition probabilities well. On the other hand, the transition probabilities estimated using the HDP prior show the kind of "smoothing" the prior was meant to encourage, where states corresponding to contexts with same suffixes have similar estimated transition probabilities.

Lastly, we plot the convergence of the entropy rate estimators with increased length of the data sequence and the associated error in Figures 2b,c. If the true depth of the model is no larger than the depth $k$ considered in the estimators, all the estimators considered should converge. We see in Figure 2c that the HDP-based entropy rate estimates converge quickly with increasing data, relative to other models.

The motivation of the hierarchical prior was to allow observations of transitions from shorter contexts to inform estimates of transitions from longer contexts. This, it was hoped, would mitigate the drop-off with larger block-size seen in block-entropy based entropy rate estimators. Figure 2d indicates that for simulated data that is indeed the case, although we do see some bias the fully Bayesian entropy rate estimator for large block lengths. The empirical Bayes and and fully Bayesian entropy rate estimators with HDP priors produce estimates that are close to the true entropy rate across a wider range of block-size.

## 7.2 Neural Data

We applied the same analysis to neural spike train data collected from primate retinal ganglion cells stimulated with binary full-field movies refreshed at 100 Hz [8]. In this case, the true transition probabilities are unknown (and indeed the process may not be exactly Markovian). However, we calculate the plug-in transition probabilities from a longer data sequence (167,000 bins) so that the estimates are approximately converged (black trace in Figure 3a), and note that transition probabilities from contexts with the same most-recent context symbols do appear to be similar. Thus the estimated transition probabilities reflect the idea that more distant context cues matter less, and the smoothing of the HDP prior appears to be appropriate for this neural data.

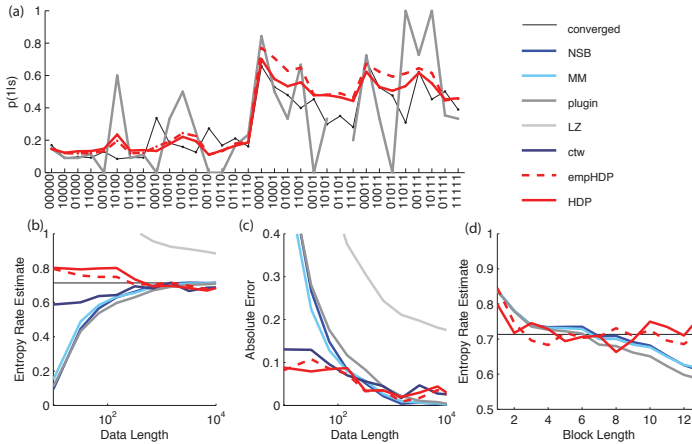

Figure 3: Comparison of estimated (a) transition probability and (b,c,d) entropy rate for neural data. The 'converged' estimates are calculated from 700s of data with 4ms bins (167,000 symbols). In (a) and (d), training data sequences are 500 symbols (2s) long. The block-based and HDP estimators in (b) and (c) use block size $k = 8$. In (b,c,d), results were averaged over 5 data sequences sampled randomly from the full dataset.

The true entropy rate is also unknown, but again we estimate it using the plugin estimator on a large data set. We again note the relatively fast convergence of $h_{HDP}$ and $h_{empHDP}$ in Figures 3b,c, and the long plateau of the estimators in Figure 3d indicating the relative stability of the HDP entropy rate estimators with respect to choice of model depth.

## 8   Discussion

We have presented two estimators of the entropy rate of a spike train or arbitrary binary sequence. The true entropy rate of a stochastic process involves consideration of infinitely long time dependencies. To make entropy rate estimation tractable, one can try to fix a maximum depth of time dependencies to be considered, but it is difficult to choose an appropriate depth that is large enough to take into account long time dependencies and small enough relative to the data at hand to avoid a severe downward bias of the estimate. We have approached this problem by modeling the data as a Markov chain and estimating transition probabilities using a hierarchical prior that links transition probabilities from longer contexts to transition probabilities from shorter contexts. This allowed us to choose a large depth even in the presence of limited data, since the structure of the prior allowed observations of transitions from shorter contexts (of which we have many instances in the data) to inform estimates of transitions from longer contexts (of which we may have only a few instances).

We presented both a fully Bayesian estimator, which allows for Bayesian confidence intervals, and an empirical Bayesian estimator, which provides computational efficiency. Both estimators show excellent performance on simulated and neural data in terms of their robustness to the choice of model depth, their accuracy on short data sequences, and their convergence with increased data. Both methods of entropy rate estimation also yield estimates of the transition probabilities when the data is modeled as a Markov chain, parameters which may be of interest in the own right as descriptive of the statistical structure and time dependencies in a spike train. Our results indicate that tools from modern Bayesian nonparametric statistics hold great promise for revealing the structure of neural spike trains despite the challenges of limited data.

## Acknowledgments

We thank V. J. Uzzell and E. J. Chichilnisky for retinal data. This work was supported by a Sloan Research Fellowship, McKnight Scholar's Award, and NSF CAREER Award IIS-1150186.

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
