[Reviews · NeurIPS 2013]

Submitted by Assigned_Reviewer_6

Reviewer's response to the rebuttal

"line 308: [10, 400] range for alpha
We chose 400 because higher values of alpha caused computational difficulties in the Gibbs sampling. This upper bound is a bit arbitrary; however, we found the exact upper limit above about 100 had little effect on the estimates of transition probabilities. Similarly, adjusting the lower bound below 60 or so had little effect. While we do hope for a more well justified hyperprior for alpha in future work, we believe our choice did not overly influence the results."

Please add this information to the paper. For someone trying to use your approach, this information is important.

Review for "Spike train entropy-rate...."

The paper introduces an entropy rate estimator for binary (neural spike) events based on a hierarchical Dirichlet process (HDP) with
binomial observation models. After surveying previous approaches and their shortcomings, the Markov chain HDP estimator is introduced, which is
designed to deal with the problem of unknown blocklength, and sparse (relative to the number of possible "phrases") observations. Two instantiations
of this estimator are developed: in the "empHDP", the entropy rates are estimated from the posterior expectations of the probabilities, whereas the "HDP"
version computes the expected entropy rate approximately via Gibbs sampling. Both estimators are compared to a range of relevant methods on simulated data,
where the authors find that their *HDP estimators exhibit a smaller underestimation bias than current related approaches. Finally, some retinal data are analyzed,
indicating that the estimator works as intended.

Clarity: the paper is well written.

Originality: the proposed method is novel in the context of spike train entropy estimation.

Significance: the paper addresses a relevant problem in computational neuroscience.

Quality: this is high quality work, I have only small number of remarks (see below).

line 128: "h_{MM,k}=H_{NSB}/k" you mean H_{MM,k} ?

line 184: formula 7: "s \in A" please say what A is.

lines 216-225: figure 1. If this is a Bayes net, then there should be observations attached to the intermediate nodes, too?

line 262-264: the unnumbered formula for the transition posterior: while this is clear to readers familiar with expectations of Beta distributions, it would improve the paper if you
said where these formulas come from.

line 269: is "..set g_{|k|}.." really a set, or a tuple? And why the "|k|" subscript? Do you mean just k?

line 284: "distributionand" -> distribution and

line 298: "..the conjugacy the beta ..." -> the conjugacy of the beta...

line 308: "...100 points on [10,400]..." why this range?

line 324-339: figure 2. One of the advantages of the Bayesian estimator is the availability of error bars, yet you do not plot them here. I would be interested to know if e.g. the underestimation of the HDP estimator
for larger blocklenghts is significant or not.

line 419: "...from shorter contexts.." remove one "."

I enjoyed reading your paper,
best wishes,
the reviewer.


Summary: A well written contribution to the spike train entropy estimation problem. Very suitable for NIPS.

Submitted by Assigned_Reviewer_7

The paper introduces a way to estimate the entropy rates (and
transition probabilities) of a Markov Model from observations, by
improving the empirical bin count through a tree-based Bayesian
prior which takes into account the progress of the probabilities
through time and assigns more similar future probabilities to
observations which differ further in the past.

The paper has some interesting ideas, and the sequential
construction of the Dirichlet priors and the extraction of the
estimates is elegant. Certainly, the results look good - but in
Fig. 3(c), the superiority of convergence of the HDP estimators is
not really clear compared to the others. The text in lines 407-409
seems to insinuate that the performance of the HDP estimators is
somehow special whereas it is quite on coherence with the other
methods. The transition probability, of course, is nicely
regularized, but stronger data is required to have an idea how good
this regularization is. Fig. 2(a), for instance shows a good match
of the high-probability transitions, but the low-probability
transitions seem not as well matched. This is slightly surprising,
as there are only two possibilities for the transition, so it is not
clear why the high transition probability values should be
regularized to a higher degree than the low transition probability
values.

nevertheless, one may wish some more reference to other
reconstruction methods, such as CSSR by Shalizi et al, suffix-tree
methods (Y. Singer et al.), observable operator methods
(Jaeger). There is a lot of emphasis on the weaknesses of pure
entropy estimation in the paper, but clearly also the structure of
the reconstruction mechanism hovers in the background which deserves
mention as your construction follows the spirit of the epsilon
machine reconstruction (Shalizi & Crutchfield) and may find relevant
generalizations in that direction.

Detailed comments:

- line 124: what's e in log(e)?

- line 128: second H_NSB has wrong subscript

- line 177: it is not clear what the bold characters mean -
throughout the paper, until quite late, I was confused whether the
new symbol appears at the *right* or the *left* end of the
string, and my confusion was caused mainly by this
illustration. Why does 010 transitionf to 101? If the new symbol
is on the right, where does the 0 at the left end disappear to? Do
we have a 2-time step memory system only here?

- line 227: similar issue - "g_0011 should be similar to g_011" -
which one is the latest symbol. Here it seems it may be the first
one. Later in the text you seem to refer to the latest symbol at
the right. Please clarify.

- line 249: "whose sample paths are each probability
distributions" - this sentence is unclear, information seems
missing; what condition do the probability distributions
associated with the paths have to fulfil?

- line 264: there is information missing here - it seems that when
you start with a fresh distribution (p_0), you still seem to
assume existence of N observations (the length of your data
string). How come you do not start with the empty string straight
away and build up your Dirichlet prior as you go through the first
N entries?

- line 269: I do not understand the structure of the posterior
predictive distribution. You state the set g_{|k|}, but I do not
see the connection between the set and the required structure of
the distribution. Reformulate. Also the rest of the paragraph
below line 272 is unclear. Please fix that, maybe it is a
consequence that I did not understand your construction of g_pr.

- line 294: I have the impression that the line contains an error:
the Beta function for the g_{0s} and the g_{1s} terms probably
need an \alpha_{|s|+1} instead of \alpha_{|s|}

- line 301: "each \alpha as fixed hyperparameters" - elide the "s"
Summary: Interesting technique; some more reference to existing process
reconstruction methods required. Unsure about expressiveness of
results. Some encouraging features, but achievements need to be more
explicit, with a few statements about comparison to the other
methods, which is not so clear, especially in 3(c).

Submitted by Assigned_Reviewer_8

The authors propose a method to estimate the entropy rate of a stationary
binary process by approximating the process by a depth-k Markov process
and then using the fact that the entropy rate of a Markov process can be calculated
analytically. So what they need to do is approximating the transition
probabilities for the Markov process.
They use a Bayesian estimator of entropy rate based on depth-k Markov model
and assume that they are given an appropriate prior.
The entropy rate estimate obtained from this approach is a good one when k is
large but the problem is by increasing k, estimating the transition probabilities
become more difficult. So the main idea of this paper is that they assume
that removing the more distant bit of the context does not change
the a priori probabilities by much. They use Beta distribution to model the prior.
It is not however explained how the parameter \alpha is estimated.

The authors also show that their method can be generalized it to non-binary data in which case
Dirichlet distribution is to be used, hence the name hierarchical
Dirichlet.
In Section 5, the authors suggest an approach to estimate the transition probabilities,
p(1|s) but why is that consistent with equation (3)?
Section 6 to perform the Gibbs sampling the posterior conditional distribution
of each g_s is needed, and it is obtained using the fact that g_s is conditionally independent
of all given its neighbors in the tree. The conditional distribution for of \alpha_is given g_s; g_0s; g_1s, is assumed to be a beta distribution, but why is that?


In Section 7 simulation result for a simulated data with known entropy
rate and neural data with an unknown entropy rate are presented and are compared
against LZ, plug in, context tree weighting, NSB, and MM. I find
the comparison somewhat unfair because because the simulated data was generated
from a Markov model. As the method is designed for Markov sources, a better
performance is not surprising. It would be
nice if the authors compared the aforementioned methods for a non-Markov data for sake of completeness even if they approach was not always the best.

Summary: Overall, The idea is good, but the execution could have improved.There are some typos here and there for instance page 5,
paragraph 2. A careful reread of the paper is recommended.
Author Feedback

Author rebuttal: We thank the reviewers for their helpful comments and suggestions.

Reviewer 1

-lines 128, 284, 298, 419
Reviewer is correct. Thank you.

-line 184
A is the set of states in the Markov chain (line 181). The formula applies to any ergodic Markov chain, so we make no assumption about what the states of the chain are. In the case of present interest, they are strings of 0s and 1s.

-line 262-664: formula for posterior transition
We appreciate the suggestion and will include and a relevant reference for the derivation.

-Figure 1
We do not attach observations to the intermediate nodes, but just use them as part of the structure of the hierarchical prior, where at the bottom level of the hierarchy sit the parameters responsible for generating the data.

-line 269
Yes, absolute value bars in the |k| subscript are unnecessary. We have g_k consisting of one scalar value for each of the 2^k binary words of length k. It matters which value goes with which word, so perhaps the ordering of a tuple is the easiest way to keep this structure intact, as the reviewer suggests.

line 308: [10, 400] range for alpha
We chose 400 because higher values of alpha caused computational difficulties in the Gibbs sampling. This upper bound is a bit arbitrary; however, we found the exact upper limit above about 100 had little effect on the estimates of transition probabilities. Similarly, adjusting the lower bound below 60 or so had little effect. While we do hope for a more well justified hyperprior for alpha in future work, we believe our choice did not overly influence the results.

Figure 2: Error bars
We did examine error bars, and consider their availability one of the advantages of our method. We ultimately omitted error bars from the figure for the sake of readability, but recognize that this left out potentially interesting information. The underestimation of the HDP estimator for larger block lengths was not significant.

Reviewer 2

-lines 128, 294, 301
Reviewer is correct. Thank you.

-Fig 3c
We agree that the superiority of the HDP in figure 3c is not clear. We will redo this figure with more data so that the relative performances of the methods are clearer.

-References to other reconstruction methods, epsilon machine reconstruction
We thank the reviewer for the suggestions, and will add these.

-line 124: log(e)
Here, e is Euler’s constant, and log(e) looks trivial, but is actually necessary because we are computing entropy in bits, and all logarithms in the paper are base 2 (line 77).

-lines 177, 227
The newest (latest in time) 0 or 1 is always rightmost. On line 177, we are explaining the transitions in a Markov model, which has finite memory. The example on line 177 is for a Markov model where states consist of 3-symbol sequences. Thus when 010 transitions to 101, a new 1 is added to the right of 010, and the leftmost 0 of 010 disappears into the past and is ‘forgotten’. Similarly on line 227, g_0011 is similar to g_011 because the contexts 0011 and 011 are identical in the 3 most recent symbols.

-line 249: sample paths
We intended just to point out that draws from the DP are probability distributions, and that the DP provides a distribution over distributions. We used the language of paths simply to clarify how the DP is in fact a stochastic process.

-line 264
N in the expression corresponding to s =\empty is correct as written. In the corresponding expression for a nonempty s, the denominator includes c_s, the number of observations of the context s, or the number of symbols preceded by context s. When the context is the empty string, we consider all N symbols to be preceded by \empty. We can include a reference to the derivation to make this clearer.

-line 269
We apologize if the notation was unclear. To specify a posterior
predictive distribution given a model of depth k, we need the
posterior values of p(1|s) for all s of length k. Line 264 gives a
method for finding these probabilities from the data. Hence, we have specified a recursive method for recovering the posterior predictive probabilities, and have an easily computable point estimate for each transition probability g_s. However, the point estimate does not provide information about the distribution of each g_s, and so while we gain speed with this method, we lose other desirable qualities, which is the point of the discussion in the next paragraph.

Reviewer 3

-Estimating the p(1|s) in section 5: why consistent with eq 3?
We agree that this is unclear. The derivation would not fit here, but we should have pointed the reader to Teh et al., 2006 and provided some intuition; we can do this in the final version of the paper

-Section 6: conditional distribution for alpha_i
We put a uniform prior on the alpha_i, and so the conditional distribution given comes from Bayes theorem and is equal to the product, over all contexts s of length i-1, of p(g_0s|g_s, alpha_i) and p(g_1s|g_s, alpha_i). We modeled both p(g_0s | g_s , alpha_i)and p(g_1s | g_s, alpha_i) as beta distributions, so the conditional probability of alpha_i is a product of beta distributions. On line 305, Beta in the denominator indicates the beta function, not the beta distribution, and it provides the normalization constant for each of the beta distributions. There is a typo here- the denominator should be squared, and the product should be over the whole fraction, not just the denominator. Perhaps this typo was the source of the confusion-we apologize.

-comparison to non-Markov data
We understand the reviewer's concern. However, we are designing the model with the task at hand in mind: we expect neural data to have the property that very distant symbols matter little. The neural data not truly Markovian, though, and so we considered performance on this data to be one test of the model’s performance on non-Markov data. We plan to redo figure 3c with more data so that the results will be clearer.